# Optimization of Flavonoid Extraction from *Xanthoceras sorbifolia* Bunge Flowers, and the Antioxidant and Antibacterial Capacity of the Extract

**DOI:** 10.3390/molecules27010113

**Published:** 2021-12-24

**Authors:** Henghui Zhang, Xiaoli Wang, Dongliang He, Dongliang Zou, Runzhu Zhao, Huifang Wang, Shuying Li, Yongping Xu, Buhailiqiemu Abudureheman

**Affiliations:** 1Department of Environment and Safety Engineering, Taiyuan Institute of Technology, Taiyuan 030008, China; teammatepolly@126.com (H.Z.); wangxl@tit.edu.cn (X.W.); hedl@tit.edu.cn (D.H.); zdlnn@163.com (D.Z.); zhaorz@tit.edu.cn (R.Z.); huifangwang2010@163.com (H.W.); 2SEM Bio-Engineering Technology Co., Ltd., Dalian 116600, China; 3School of Bioengineering, Dalian University of Technology, Dalian 116024, China; 4School of Food Science and Engineering, Xinjiang Institute of Technology, Aksu 843100, China

**Keywords:** *Xanthoceras sorbifolia* Bunge, total flavonoids, ultrasound-assisted extraction, response surface methodology, antioxidant activity, antibacterial activity

## Abstract

In the present work, the extraction process of total flavonoids (TFs) from *X. sorbifolia* flowers by ultrasound-assisted extraction was optimized under the response surface methodology (RSM) on the basis of single-factor experiments. The optimal extraction conditions were as follows: ethanol concentration of 80%, solid–liquid ratio of 1:37 (g/mL), temperature of 84 °C, and extraction time of 1 h. Under the optimized conditions, the extraction yield of the TFs was 3.956 ± 0.04%. The radical scavenging capacities of TFs against 2,2-diphenyl-1-picrylhydrazyl (DPPH) and 2,2-azino-bis(3-ethylbenzothiazoline-6-sulphonic acid) (ABTS) were much greater than that of rutin. The results of antibacterial experiments indicated that the TFs displayed strong inhibitory activities on *E. coli*, *S. aureus* and *Bacillus subtilis*. Therefore, *X. sorbifolia* flowers can be used as a novel source of natural flavonoids, and the TFs have potential applications as natural antioxidants or antibacterial agents in the food and pharmaceutical industries.

## 1. Introduction

*Xanthoceras sorbifolia* Bunge is a monotype species endemic to China, belonging to the genus Sapindaceae [1]. It is a woody oil crop with a life span of more than 200 years, and it is also a homologous medicine and food plant with high economic value [2,3]. *Xanthoceras sorbifolia* Bunge has the advantages of early fructification, high seed oil content, drought resistance, salt and alkali resistance, large suitable area, easy reproduction, etc. All parts of *Xanthoceras sorbifolia* Bunge can be used; however, the kernel is still the most widely used part. In addition to the rich linoleic acid and linolenic acid in the oil of *X. sorbifolia* seeds, there is a range of flavonoids, polysaccharides, saponins, and other bioactive components in the flowers, leaves, and stem of *X. sorbifolia* [4].

In previous studies, the main research focused on the development and utilization of *X. sorbifolia* oil, and not enough research has been conducted on the other active ingredients [5,6,7]. However, besides *X. sorbifolia* fruits, its flowers, leaves, and stem can all be used as medicine in traditional Chinese medicine, among which flavonoids are one of the main molecular components of its biological function [8].

Flavonoids are a group of secondary metabolites founds in many plants, whose parent nucleus has the structure of 2-phenylchromone [9,10]. As natural compounds, flavonoids have attracted much attention because of their specific biological activities such as preventing cell senescence, resisting bacterial infection, and improving the body’s immunity [11,12]. Many medicinal mechanisms of flavonoids were closely linked to antioxidant and antibacterial activities [13,14,15].

Research on the flavonoids of the *X. sorbifolia* flowers is rare. The aim of this study was to optimize the extraction process of total flavonoid from the flowers of *X. sorbifolia* and evaluate the antioxidant and antimicrobial activities of flavonoids. This research provided reliable data for studying the flavonoids of *X. sorbifolia* flowers for applications in the food and pharmaceutical industries.

## 2. Materials and Methods

### 2.1. Chemicals and Bacterial Strains

A standard rutin sample, a standard vitamin C (Vc) sample, and a standard tea polyphenol sample were obtained from Aladdin Reagent Co., Ltd. (Shanghai, China). 2,2-diphenyl-1-picrylhydrazyl (DPPH) and 2,2-azino-bis (3-ethylbenzothiazoline-6-sulfonic acid) (ABTS) were obtained from Sigma-Aldrich (St. Louis, MO, USA). NaOH, NaNO_2_, AlNO_3_, ethanol, phosphoric acid, and concentrated hydrochloric acid were obtained from Sinopharm Chemical Reagents Co. Ltd. (Shanghai, China). Standard Escherichia coli strain C83902, Staphylococcus aureus strain ATCC 25923, and Bacillus subtilis strain WL2 were obtained from the China Institute of Veterinary Drug Control (Beijing, China).

### 2.2. Plant Material

The flowers of *X. sorbifolia* Bunge were collected from Chaoyang City, Liaoning Province, China and identified by professor Xiangping Pei from the School of Traditional Chinese Medicine, Shanxi University of Chinese Medicine. Then, the flowers were dried under natural ventilation to constant weight, powdered, and immediately stored in a dry container for further use.

### 2.3. Measurement of Total Flavonoids Content

The contents of total flavonoids were measured by aluminum nitrate colorimetry using rutin as standard with slight modifications [16]. Firstly, 0.15 mL of NaNO_2_ (5%, *w*/*v*) was added into the sample solution (0.5 mL). After 6 min, 0.15 mL of (10%, *w*/*v*) AlNO_3_ was added into the mixture. Another 6 min later, 2 mL of NaOH (1 M) was added. The mixture was then made up to a final volume of 5 mL via the addition of ultrapure water. Finally, the complete mixture was shaken and incubated for 20 min until a sufficient reaction occurred, and the absorbance was immediately read at 510 nm. A standard curve was plotted using rutin solutions with different concentration gradients (0–60 μg/mL), and the TFs were calculated according to the regression equation: y = 8.517x + 0.006 (R^2^ = 0.995), where y is the absorbance and x is the content (μg/mL). The yield of TFs was calculated by the equation given below:(1)$$\mathrm{Extraction}\;\mathrm{yield}\;(\mathrm{mg}/g)=\frac{C\times V}{W}$$
where *C* represents the TFs (mg/mL); *V* represents the total volume of the extract (mL); and *W* represents the weight of the *X. sorbifolia* Bunge flower powder (g).

### 2.4. Optimization of Extraction of TFs from Xanthoceras sorbifolia Flowers

#### 2.4.1. Single-Factor Experiments

The ultrasound-assisted extraction method (UAE, 600 W) was used to extract the TFs from *X. sorbifolia* flowers [17]. Based on the pre-experiment, four factors were selected as independent variables to study their effects on UAE, which were ethanol concentration, solid–liquid ratio, extraction temperature, and extraction time. First, to study the impact of ethanol concentration on the yield of TFs, 1.0 g of *X. sorbifolia* Bunge flower powders were soaked in a series of concentrations of ethanol solvents (50%, 60%, 70%, 80%, and 90%, *v*/*v*), the solid–liquid ratio was set at 1:30 (g/mL), and the extraction temperature and extraction time were set at 70 °C and 1 h, respectively [18]. Then, to investigate the effect of extraction temperature, different extraction temperatures (50, 60, 70, 80, and 90 °C) were arranged under the conditions of 80% ethanol concentration, 1:30 (g/mL) solid–liquid ratio, and 1 h extraction time [19]. Subsequently, multiple solid–liquid ratios (1:10, 1:20, 1:30, 1:40, and 1:50, g/mL) were tested under the conditions of 80% ethanol concentration, 70 °C extraction temperature, and 1 h extraction time to evaluate the effects of the solid–liquid ratios on the TFs’ yield. Finally, to choose the optimum extraction time, different extraction time (0.5, 1.0, 1.5, 2.0 and 2.5 h) were studied under the following conditions: ethanol concentration 80%, solid–liquid ratio 1:40 (g/mL), and extraction temperature 70 °C [20]. At the end of each extraction, the supernatant was collected after centrifugation (5000 rpm, 4 min), and the residue was re-extracted twice more under the same condition. The combined filtrate was then concentrated to a specified volume and the final extract was stored for further analysis [21].

#### 2.4.2. Response Surface Methodology (RSM) Design

On the basis of single factor experimental results, the extraction of TFs from *X. sorbifolia* Bunge flower was optimized using the response surface methodology (RSM) [22,23]. In this study, the Box–Behnken design method provided by Design Expert 11 (Stat-Ease Inc., Minneapolis, MN, USA) was selected to guide the experimental design with four variables (extraction temperature, ethanol concentration, solid–liquid ratio, and extraction time) at three levels, whose response value was the extraction yield of the TFs (Y) [24]. The Box–Behnken design method with four factors and three levels consisted of 29 randomized experiments [25]. According to the experimental results of each trial, a quadratic equation model would be fitted to calculate the extraction yield and assess the importance of the factors and their interactions in the extraction process. The design of experiment is shown in Table 1.

### 2.5. Evaluation of Antioxidant Activity

In recent years, many methods have been used to estimate the antioxidant activity of antioxidant candidates [26]. The commonly used in vitro assays involved DPPH, ABTS, hydroxide radical scavenging activity, ferric ion reducing antioxidant power (FRAP), etc. [27]. In this work, the DPPH and ABTS radical scavenging activities were selected for the rapid assessment of the potential antioxidant activity of the TFs from *X. sorbifolia* Bunge flower, providing a basis for further research on antioxidant activity in vivo [28].

#### 2.5.1. Assay of DPPH• Radical Scavenging Activity

The DPPH• radical scavenging capacity of the TF extracts from *X. sorbifolia* Bunge flower was tested based on the method reported by Farias et al. 2020 with slight modification [28]. Briefly, 0.1 mL of the TF extract samples with different concentrations (5, 10, 15, 20 and 25 μg/mL) were mixed with 0.9 mL of the the DPPH solution (60 μmol/L soluble in ethanol). After blending, the mixture was placed in darkness for 30 min. Then, absorbance of the mixture at 518 nm was recorded immediately with ethanol as a reference [29]. Ascorbic acid and rutin were used as positive controls in this study. The percentage of DPPH• radical scavenging was calculated using the following equation:(2)$$\mathrm{DPPH}\;\mathrm{radical}\;\mathrm{scavenging}\;\mathrm{activity}\;(\%)=(1-\frac{A_{S}-A_{r}}{A_{0}})\times 100$$
where *A_s_* represents the absorbance of sample solution after reacting with DPPH solution, *A_r_* represents the absorbance of sample solution with reference (absolute ethanol), and *A_0_* represents the absorbance of ultrapure water with DPPH solution. Each test was repeated three times. The EC_50_ was defined as the concentration of the TF extract, which achieved the effect of scavenging 50% of the initial DHHP free radical.

#### 2.5.2. Assay of ABTS•+ Radical Scavenging Activity

The ABTS•+ radical scavenging activity assay method developed by Lou was adopted with minor modifications [30]. Primarily, in order to produce abundant ABTS free radicals, the ABTS stock solution was prepared by mixing the same volume of refined ABTS solution (14 mM) with K_2_S_2_O_8_ solution (4.9 mM) for 12 to 16 h at room temperature in the dark. Then, the ABTS solution was diluted to the working state by adding 70% ethanol, which required an absorbance of 0.70 ± 0.02 at 734 nm. The ABTS working solution needed to be applied on the spot. A quantity of 300 μL of the ABTS working solution was mixed with 100 μL of the TF extract solution at different concentrations (5, 10, 15, 20, and 25 μg/mL). After reaction for 6 min at room temperature in the dark, the absorbance of mixture at 734 nm was recorded [9]. The same procedure was used to test ascorbic acid and rutin as positive controls in this experiment. The percentage of ABTS•+ radical scavenging was calculated using the Equation (3):(3)$$\mathrm{ABTS}\;\mathrm{radical}\;\mathrm{scavenging}\;\mathrm{activity}\;(\%)=(1-\frac{A_{S}-A_{r}}{A_{0}})\times 100$$
where *A_s_* represents the absorbance of the samples after reacting with ABTS working solution, *A_r_* represents the absorbance of the samples after adding 70% ethanol, and *A_0_* represents the absorbance of ultrapure water with ABTS working solution. Each test was repeated three times.

### 2.6. Evaluation of Antibacterial Activity

According to the implementation standards of antimicrobial susceptibility test recommended by the National Committee on Clinical and Laboratory Standards (NCCLS), the Oxford cup method was used to detect the inhibition of bacterial growth of TF extracts from *X. sorbifolia* Bunge flower [31]. Gram-negative Escherichia coli bacteria, and Gram-positive Staphylococcus aureus and Bacillus subtilis bacteria, were selected as test strains. Bacteria in the logarithmic growth phase was uniformly diluted with sterilized saline to a concentration of 10^6^ CFU/mL until it was ready for use, following McMillan’s turbidimetric method [32]. The Oxford cups were placed on MH agar medium evenly coated with diluted bacterial suspension, and then 100 μL of TF extract samples with different concentrations were injected into the cups [33]. After 24 h incubation in 37 °C constant-temperature incubator, the diameters of the inhibition zones on the medium were measured with a vernier caliper for analysis [34]. Tea polyphenols and 95% ethanol were used as positive controls, and ultrapure water was also used as a blank control. Three replicates were set for each sample.

### 2.7. Statistical Analysis

All experiments were repeated at least 3 times independently, and data were expressed as means ± SD for each experiment. An analysis of variance (ANOVA) was used and differences in the average extraction amounts of total flavonoids were analyzed using the F-test. The results of RSM were tested using Design-Expert 11 (Stat-Ease Inc., Minneapolis, MN, USA). The radical scavenging and antibacterial activity analyses were performed with the GraphPad Prism 5.0 program (GraphPad Software Inc., La Jolla, CA, USA). *p* ≤ 0.05 was considered statistically significant.

## 3. Results

### 3.1. Single-Factor Experiments

#### 3.1.1. Effect of Ethanol Concentration on the TFs’ Yield

Flavonoids are generally insoluble in water and soluble in organic solvents, and thus, ethanol is one of the best solvents. For this reason, the concentration of ethanol is an important factor that affects the extraction yield of TFs from *X. sorbifolia* Bunge flower. The effects of ethanol solutions of different concentrations on the extraction yield of total flavonoids are shown in Figure 1a. As shown, the yield of the TFs rose with the increasing of the ethanol concentration from 50 to 80%. However, when the concentration of ethanol exceeded 80%, the extraction yield of the TFs began to decline with the increase in concentration, which was probably due to the structural change of flavonoids in the extract caused by the high concentration of ethanol. Therefore, 80% ethanol was selected as the optimal extraction solvent in the single factor experiment.

#### 3.1.2. Effect of Extraction Temperature on the TFs’ Yield

Extraction temperature is undoubtedly a remarkable factor in the extraction of bioactive substances. As shown in Figure 1b, the initial increment of the extraction temperature (from 50 to 70 °C) resulted in an obvious improvement in the TFs’ yield. When the temperature rose from 70 °C to 80 °C, the extraction yield increased slowly. Eventually, the TFs’ yield began to decrease with the further rise in extraction temperature above 80 °C. Studies have shown that the increasing of the temperature is conducive to the separation of the extracted flavonoids from the original system, whereas excessive temperatures easily deactivate them. Thus, 80 °C was chosen as the proper extraction temperature.

#### 3.1.3. Effect of Solid–Liquid Ratio on the TFs’ Yield

In this work, the influence of diverse solid–liquid ratio on the TFs’ yield was studied and the results are shown in Figure 1c. When the quantity of ethanol solvent was low, the extraction yield of flavonoids increased with the decreasing of the solid–liquid ratio. Until the maximum extraction yield of about 4.5 mg/g was reached at the solid–liquid ratio of 1:40, the TFs’ yield exhibited a trend of stabilization or even slight decline with the increasing of the ethanol solution. The possible reason for this result was that adequate contact between the solid and liquid could have promoted material exchange and improved the extraction amount of flavonoids until the maximum value was reached. In consideration of efficient material use, 1:40 (g/mL) was regarded as the best solid–liquid ratio.

#### 3.1.4. Effect of Extraction Time on the TFs’ Yield

The effect of extraction time (0.5 to 2.5 h) on the TFs’ yield is revealed in Figure 1d. At the beginning, along with the extension of extraction time (from 0.5 to 1.5 h), the TFs’ yield improved. However, when the extraction time exceeded 1.5 h, the extraction yield result dropped from 5.03 mg/g at 1.5 h to 4.40 mg/g at 2.5 h. In general, the extension of the extraction time contributes to the extraction process, but excess heating time at a higher temperature would nevertheless cause the degradation of some flavonoids. Hence, the optimal extraction time was set at 1.5 h.

### 3.2. Optimization of Extraction Process by RSM

#### 3.2.1. Model Fitting and Statistical Analysis

According to Box–Behnken design method, the experimental data obtained using the response surface methodology are given in Table 1. By means of statistical regression analysis, a quadratic polynomial regression equation of the RSM test results was fitted for the extraction yield of TFs, as shown below:Y% = 3.88 + 0.27A + 0.19B + 0.089C−0.12D−0.014AB + 0.062AC − 0.05AD
+ 0.053BC − 0.098BD + 0.007CD − 0.16A^2^ − 0.18B^2^ − 0.17C^2^ − 0.19D^2^(4)
where Y is the predicted value of TF yield, and A, B, C, and D are the extraction temperature, ethanol concentration, solid–liquid ratio, and extraction time, respectively.

ANOVA was used to evaluate the fitness of the regression model, and the results are shown in Table 2. The *F*-value (10.09) and *p*-value (<0.0001) of the regression model indicated that the fitting model was extremely significant. However, the lack of fit was not significant, which indicated that further optimization experiments were not required. Furthermore, the decisive coefficient (R^2^) of the model was 0.9809, which indicated that the model could satisfactorily fit the variability of the response value. The adjusted R^2^ value of 0.9712 indicated that the changes of 95.12% in the response value model came from the selected variables, implying that the four variables selected had great influence on the variation of response value. According to the F value, the order of influence of the four factors on the TFs’ yield was A (extraction temperature), B (ethanol concentration), D (extraction time), and C (solid–liquid ratio). The linear terms A and B in the model showed extremely significant effects of the extraction yield (*p* < 0.001). The linear term D, and the quadratic terms A^2^, B^2^, C^2^, and D^2^, showed highly significant effects (*p* < 0.01). The linear term C had a significant effect (*p* < 0.05).

#### 3.2.2. Analysis of Interaction between Factors

The three-dimensional (3D) response surface images of the interaction of A (extraction temperature), B (ethanol concentration), C (solid–liquid ratio), and D (extraction time) are shown in Figure 2. The shapes of the 3D response surface plots reflect the influence of this interaction on the response value. The ellipse or saddle shape of a 3D surface plot usually indicates that the interaction between two independent variables on the response value is significant. However, a flat surface indicates that the interaction between the two factors on the response value is not obvious. Figure 2 shows that the extraction temperature and ethanol concentration corresponded to a steep curve, which implied that these two variables had the greatest influence on the extraction yield of TFs. The contour plots under 3D surface of ethanol concentration and solid–liquid ratio, e=thanol concentration and extraction time, and extraction temperature and solid–liquid ratio, were oval, which indicated that the interactions had a significant influence on the extraction yield of TFs from *X. sorbifolia* flowers.

Among the six response surface plots, the projection of the response surface plot of extraction temperature and ethanol concentration shown in Figure 2a is the closest to a circle, which means the interaction between the two factors was minimal. As shown in Figure 2e, the projection was most elliptical, which means that the interaction between ethanol concentration and extraction time was maximal.

#### 3.2.3. Validation of the Predictive Model

The optimal conditions for extraction of total flavonoids from *X. sorbifolia* flowers predicted by RSM model were as follows: ethanol concentration of 80.82%, extraction temperature of 84.23 °C, solid–liquid ratio of 1:37.39 (g/mL), and extraction time of 1.1 h. Under these conditions, the theoretical extraction yield of TFs was 4.144 ± 0.05%. Then, the optimal conditions were modified so that the ethanol concentration was 80%, the extraction temperature was 84 °C, the solid–liquid ratio was 1:37 (g/mL), and the extraction time was 1 h. Subsequently, the confirmatory experiments were arranged under these conditions to verify the accuracy of the prediction model, and the extraction yield result after repeated tests was 3.956 ± 0.04%, which was close to the theoretical value. The results indicated that the regression model was well-fitted and could be used to optimize the extraction process of TFs from *X. sorbifolia* flowers.

### 3.3. Antioxidant Activity

#### 3.3.1. DPPH• Radical Scavenging Capacity

The principle of the current methods used for the in vitro determination of the antioxidant capacity of antioxidants is mainly based on their activities in inhibiting lipid oxidation and degradation, scavenging free radicals, inhibiting pro-oxidants, and reducing pro-oxidant capacity. Among them, DPPH radical scavenging activity assay is one of the commonly used methods to detect the antioxidant activity of natural products, and the testing results for the TF extract from *X. sorbifolia* flowers are shown in Figure 3. It was found that the free radical scavenging activities of the TFs extract increased with the increasing of the extract concentration, and showed a linear relationship within a certain range (R^2^ = 0.994). Compared with rutin and vitamin C, the DPPH radical scavenging capacity of the three groups at the same concentration was in the order of vitamin C, TF extract, and rutin, respectively. The EC_50_ value of DPPH radical scavenging rate of the TF extract was 10.69 μg/mL, 49.20 μg/mL for rutin, and 3.82 μg/mL for vitamin C (Table 3). The results indicated that the TF extract from *X. sorbifolia* flowers exhibited a good effect on the DPPH radical scavenging activity.

#### 3.3.2. ABTS•+ Radical Scavenging Capacity

The ABTS•+ radical scavenging capacity results for the TF extract from *X. sorbifolia* flowers, obtained at different concentrations, are shown in Figure 4. As shown, the increase in the concentration of the TF extract could have contributed to the improvement in the ABTS•+ radical scavenging activity. As controls, rutin and vitamin C showed the same trend in that the concentration and free radical scavenging ability increased linearly. Furthermore, the EC_50_ value of the ABTS free radical scavenging rate for the TF extract, rutin, and vitamin C were 14.23, 46.17, and 1.59 μg/mL, respectively, implying that the TF extract from *X. sorbifolia* flowers was an adequate antioxidant of the scavenging ABTS free radical.

### 3.4. Antibacterial Activity

In addition to their antioxidant activity, plant polyphenols, including flavonoids, often show antimicrobial activity. In this study, the antibacterial activities of TFs from *X. sorbifolia* flower and tea polyphenols, against three kinds of microbes, were proven to be anabatic with the increase in concentration. According to statistical analysis, the difference between the experimental groups and the blank control were all extremely significant (*p* < 0.001). Thus, following the principles of NCCLS, the susceptibility tests of TFs from the *X. sorbifolia* flower, at different concentrations, were regarded as intermediate (I level, inhibition zone diameter between 12 to 18 mm). Among the three species of bacteria, the inhibitory activity of the TFs extract at any concentration against *E. coli* was the best, followed by *S. aureus*, and the inhibitory activity against *Bacillus subtilis* was relatively weak (Figure 5). Against *E. coli*, the inhibitory activity the TFs extract was negligibly stronger than that of the tea polyphenol extract at low and medium concentrations, without significant difference, yet it was significantly stronger at high concentration with the effect being nearly equivalent to that of 95% ethanol (Figure 5a). Referring to *S. aureus* and *Bacillus subtilis*, the TFs extract exhibited indistinctively weaker effect on the inhibition of growth than tea polyphenols at the concentrations of 2.1 and 4.2 mg/mL. However, the inhibition zone diameter of the TF extract at 8.5 mg/mL against *S. aureus* and *Bacillus subtilis* became slightly larger than that of the tea polyphenol extract, with no significant difference from 95% ethanol as well (Figure 5b,c).

## 4. Conclusions

In this study, we explored the extraction process of total flavonoids from *X. sorbifolia* flowers by UAE and optimized the conditions using the response surface methodology. The optimal extraction conditions for TFs were modified as follows: ethanol concentration of 80%, solid–liquid ratio of 1:37 (g/mL), temperature of 84 °C, and extraction time of 1 h. Under these conditions, the extraction yield of TFs was 3.956 ± 0.04%. Then, the total flavonoids extracted from the flowers of *X. sorbifolia* showed positive effects on scavenging DPPH• and ABTS•+ Radicals, which were much greater than that of the rutin control. Moreover, the results of antibacterial experiments indicated that the TFs extracted from *X. sorbifolia* flowers displayed strong inhibitory activities on *E. coli*, *S. aureus* and Bacillus subtilis, among which the anti-*E. coli* activity was the strongest. At the concentration of 8.5 mg/mL, the inhibitory effect of TFs extract on *E. coli* was significantly stronger than that of tea polyphenols, and comparable to 95% ethanol. In summary, the present work explored the comprehensive development of *X. sorbifolia* and pointed out potential applications of total flavonoids in the flowers of *X. sorbifolia* as novel sources of natural antioxidants or antibacterial agents in the food and pharmaceutical industries.

## Figures and Tables

**Figure 1 molecules-27-00113-f001:**
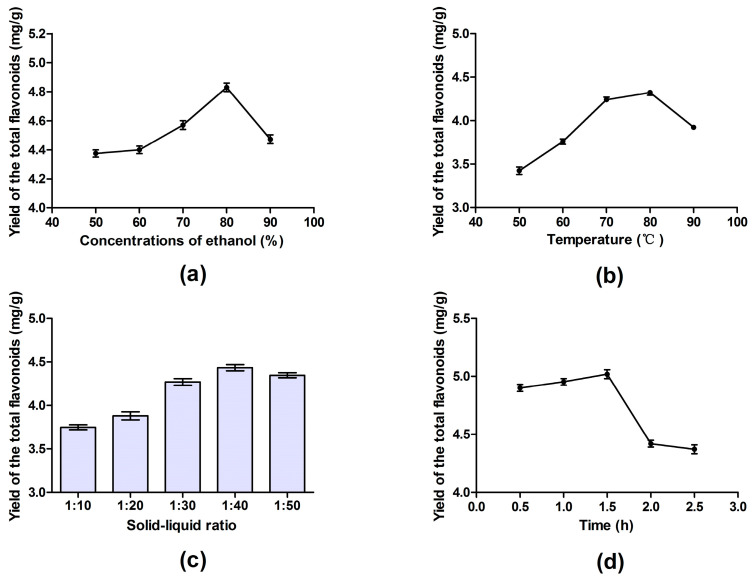
Single factor experiment results for the extraction yield of total flavonoids from the flower of *Xanthoceras sorbifolia*: (**a**) effect of ethanol concentration on the yield of TFs; (**b**) effect of extraction temperature on the yield of TFs; (**c**) effect of solid–liquid ratio on the yield of TFs; (**d**) effect of extraction time on the yield of TFs. The results are shown as mean ± SD.

**Figure 2 molecules-27-00113-f002:**
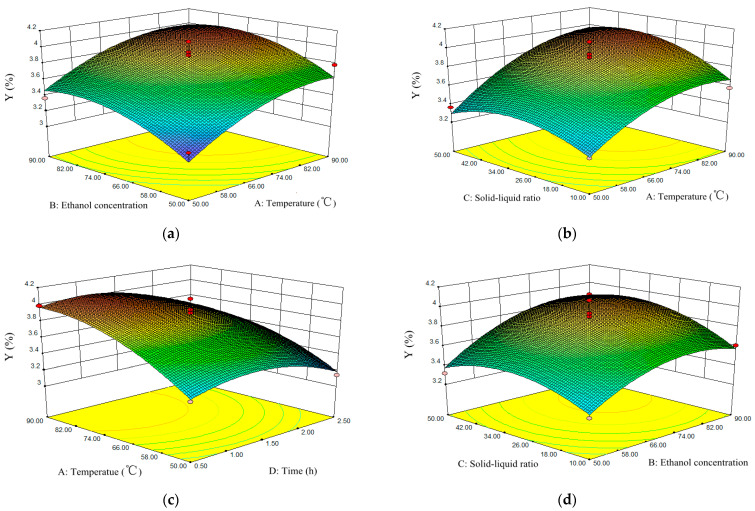
The interaction between variables of extracted TFs showed by response surface plots: (**a**) extraction temperature and ethanol concentration; (**b**) extraction temperature and ethanol and solid–liquid ratio; (**c**) extraction temperature and time; (**d**) ethanol concentration and solid–liquid ratio; (**e**) ethanol concentration and extraction time; (**f**) solid–liquid ratio and extraction time.

**Figure 3 molecules-27-00113-f003:**
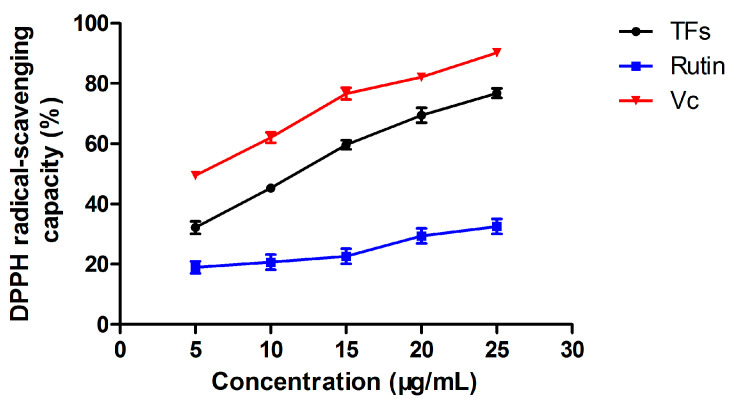
The DPPH radical scavenging capacity of TFs extracted from the flowers of *X. sorbifolia* Bunge compared with rutin and Vc. The experiment was repeated three times, and values are given as mean ± SD (n = 3).

**Figure 4 molecules-27-00113-f004:**
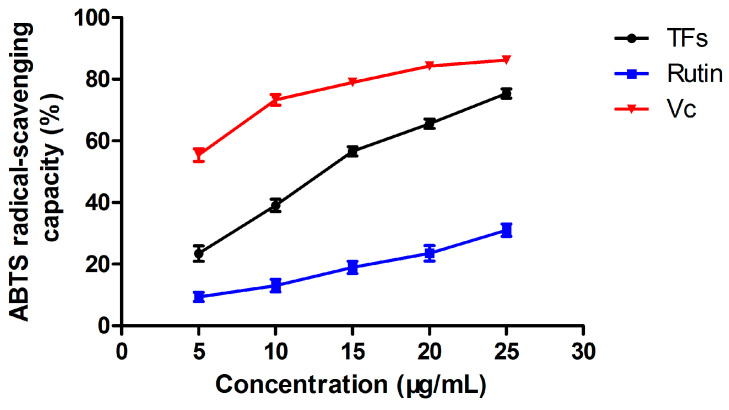
The ABTS radical scavenging capacity of TFs extracted from the flowers of *X. sorbifolia* Bunge compared with rutin and Vc. The experiment was repeated three times, and values are given as mean ± SD (n = 3).

**Figure 5 molecules-27-00113-f005:**
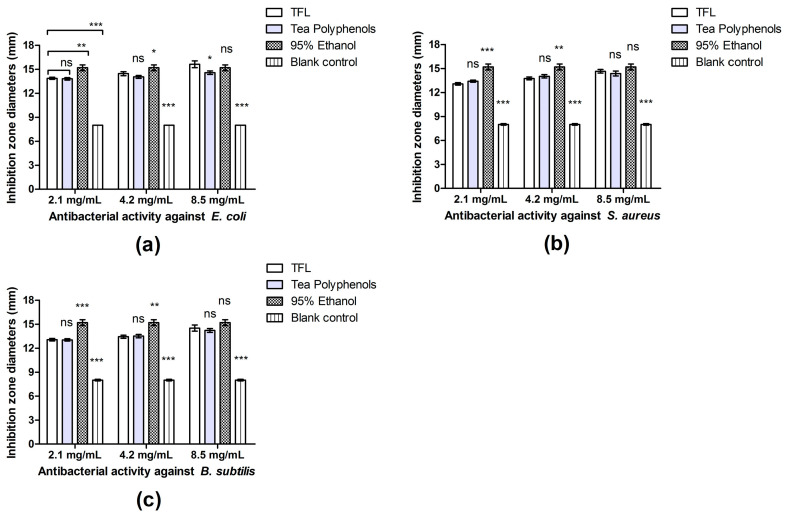
The antibacterial activities of TFs extracted from the flowers of *X. sorbifolia* Bunge against *E. coli*, *S. aureus* and *B. subtilis* at different concentrations: (**a**) sizes of inhibition zones against *E. coli* at low, medium and high concentrations compared with tea polyphenols, 95% ethyl alcohol and blank control; (**b**) sizes of inhibition zones against S. aureus at low, medium and high concentrations compared with controls; (**c**) sizes of inhibition zones against B. subtilis at low, medium and high concentrations compared with controls. Values are given as mean ± SD (n = 3). *** indicates that the difference is extremely significant (*p* < 0.001) between the given groups. ** indicates that the difference is highly significant (*p* < 0.01). * indicates that the difference is significant (*p* < 0.05). ns indicates that no significant difference exists.

**Table 1 molecules-27-00113-t001:** Box–Behnken design matrix and experimental results.

NO.	A Temperature (°C)	B Ethanol Concentration (%)	C Solid–Liquid Ratio (g/mL)	D Time (h)	Yield (mg/g)
1	50 (−1)	90 (+1)	1:30 (0)	1.5 (0)	3.364
2	90 (+1)	70 (0)	1:10 (−1)	1.5 (0)	3.575
3	50 (−1)	50 (−1)	1:30 (0)	1.5 (0)	3.165
4	50 (−1)	70 (0)	1:30 (0)	0.5 (−1)	3.294
5	50 (−1)	70 (0)	1:50 (+1)	1.5 (0)	3.364
6	70 (0)	50 (−1)	1:30 (0)	0.5 (−1)	3.188
7	90 (+1)	70 (0)	1:30 (0)	2.5 (+1)	3.634
8	70 (0)	70 (0)	1:10 (−1)	0.5 (−1)	3.669
9	50(−1)	70 (0)	1:30 (0)	2.5 (+1)	3.141
10	70 (0)	70 (0)	1:50 (+1)	0.5 (−1)	3.786
11	70 (0)	90 (+1)	1:10 (−1)	1.5 (0)	3.610
12	70 (0)	50 (−1)	1:30 (0)	2.5 (+1)	3.294
13	70 (0)	70 (0)	1:30 (0)	1.5 (0)	3.692
14	70 (0)	90 (+1)	1:30 (0)	2.5 (+1)	3.681
15	70 (0)	70 (0)	1:30 (0)	1.5 (0)	3.939
16	70 (0)	90 (+1)	1:30 (0)	0.5 (−1)	3.904
17	70 (0)	70 (0)	1:10 (−1)	2.5 (+1)	3.294
18	90 (+1)	50 (−1)	1:30 (0)	1.5 (0)	3.786
19	90 (+1)	70 (0)	1:30 (0)	0.5 (−1)	3.986
20	70 (0)	70 (0)	1:30 (0)	1.5 (0)	3.786
21	90 (+1)	70 (0)	1:50 (+1)	1.5 (0)	3.951
22	70 (0)	70 (0)	1:30 (0)	1.5 (0)	4.068
23	70 (0)	70 (0)	1:30 (0)	1.5 (0)	3.904
24	90 (+1)	90 (+1)	1:30 (0)	1.5 (0)	3.927
25	50 (−1)	70 (0)	1:10 (−1)	1.5 (0)	3.235
26	70 (0)	90 (+1)	1:50 (+1)	1.5 (0)	3.868
27	70 (0)	50 (−1)	1:10 (−1)	1.5 (0)	3.270
28	70 (0)	50 (−1)	1:50 (+1)	1.5 (0)	3.317
29	70 (0)	70 (0)	1:50 (+1)	2.5 (+1)	3.493

**Table 2 molecules-27-00113-t002:** The analysis of variance of the regression model.

Source	Sum of Squares	Df	Mean Square	*F*-Value	*p*-Value	Significance
Model	2.18	14	0.16	10.09	< 0.0001	***
A	0.91	1	0.91	58.68	< 0.0001	***
B	0.43	1	0.43	27.86	0.0001	***
C	0.096	1	0.096	6.21	0.0259	*
D	0.16	1	0.16	10.69	0.0056	**
AB	0.000841	1	0.00084	1.055	0.8188	
AC	0.015	1	0.015	0.99	0.3370	
AD	0.0099	1	0.0099	0.64	0.4365	
BC	0.011	1	0.011	0.72	0.4100	
BD	0.000841	1	0.00084	1.055	0.8188	
CD	0.001964	1	0.00196	0.013	0.9119	
A^2^	0.18	1	0.18	11.37	0.0046	**
B^2^	0.21	1	0.21	13.37	0.0026	**
C^2^	0.19	1	0.19	12.14	0.0036	**
D^2^	0.23	1	0.23	14.60	0.0019	**
Residual	0.22	14	0.015			
Lack of fit	0.13	10	0.013	0.63	0.7454	
Pure error	0.084	4	0.021			
Cor total	2.40	28				
R^2^ = 0.9809						
R^2^_Adj_ = 0.9712						

*** indicates that the difference is extremely significant (*p* < 0.001). ** indicates that the difference is highly significant (*p* < 0.01). * indicates that the difference is significant (*p* < 0.05). Df indicates the degree of freedom.

**Table 3 molecules-27-00113-t003:** The curve of the EC_50_ values of scavenging DPPH/ABTS radicals with antioxidants.

Indicators	Antioxidants	R^2^ of Linear Fit	EC_50_ (μg/mL)
DPPH•	TFs	0.984	10.69
	Rutin	0.992	49.20
	Vc	0.986	3.82
ABTS•+	TFs	0.988	14.23
	Rutin	0.991	46.17
	Vc	0.984	1.59

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
