# Peer review of "Optimization of Flavonoid Extraction from Xanthoceras sorbifolia Bunge Flowers, and the Antioxidant and Antibacterial Capacity of the Extract"

_molecules, 2021, doi:10.3390/molecules27010113_

Round 1
Reviewer 1 Report
The manuscript is suitable for publication after minor revisions
Reviewer Comments
- The manuscript entitled " Optimization of Extraction Process, Antioxidant and Antibacterial
Capacity of Total Flavonoids from the Flowers of Xanthoceras Sorbifolia Bunge " reports the evaluation of total flavonoids content from the X. Sorbifolia Bunge Flowers, their antioxidant and antibacterial effects, after optimization with the methodology of experimental designs (29 experiments) the four influencing factors on the ultrasound-assisted extraction process (T°, ethanol-concentration, solid-liquid ratio and time-effect).
In order to quantify the total flavonoids content as well as the EC50, the following standards were used: rutin and ascorbic acid.
- The structure of the manuscript is well organized and logic.
- The results section is clear and comprehensible, well stated and discussed.
- The chemical reagents used as well as the elucidation method applied are suitable for this type of study.
- The authors have determined important optimal conditions of ultrasound-assisted extraction process in X. Sorbifolia Bunge Flowers (T°= 84 °C, ethanol-concentration: 80%, solid-liquid ratio:1:37 g/ml, time duration: 1h).
- Interesting data were obtained during the study of antibacterial activities (flavonoids of flowers ethanolic extracts) which are closely neighboring to those of compared tea polyphenols.
- The reviewer regrets the absence of HPLC or HPLC/MS analyses which could provide deep information on the qualitative and quantitative chemical composition related to each experiment.
- The referencing style is compatible with the template and style of the journal.
- The language is acceptable in general, some minor corrections are required in order to improve the quality of the manuscript (please, see the comments below).
In Abstract part:
- Page 1, line 14: to develop the acronym UAE i.e. ultrasound-assisted extraction.
- Page 1, line 18: Please, change the alphabets (a, a- and b) by the following numbers (2,2- and 1-) in the nomenclature of the DPPH i.e. ‘‘2,2-diphenyl 1-picrylhydrazyl’’ instead of ‘‘a,a-diphenyl b-picrylhydrazyl’’.
- Page 1, line 17: What do you means with “TFL”? Is it “TFL” or “TFC : Total Flavonoids Content”. The same questions related to page 2, lines 79 and 87; page 3, lines 96, 104, 109, 119, 122, 124, 134; page 4, lines 144, 157, 179, 182, 184, 186; and in page 5 to page 11.
- Page 1, line 24 (keywords): Please add “ultrasound-assisted extraction”.
In manuscript text:
- Page 2, line 46: What do you means with “…rom…” ? In order to keep your text homogeneous, please discard this word.
- Page 2, line 76: What do you means with “TFY”? To develop the acronym given.
- Page 3, line 123: “…by Farias with…” change it with “…by Farias et al. 2020 with…”
- Page 3, lines 131, 132 and 133: Respect the writing of the following indices related to As, Ar and A0 i.e. “…As, …Ar and A0…”, they must be conform to the equation (2) given in page 3, line 130.
- Page 3, line 137: “…from Lou …[32]”. The author's name “Lou” does not comply with the reference [32] given in page 13. To adjust
- Page 4, lines 150 and 151: Respect the writing of the following indices relating to As, Ar and A0 i.e. “…As, …Ar and A0…”, they must be conform to the equation (3) given in page 4, line 149.
- Page 6, line 230: For more clarity to readers, the following acronym “Y …” must be introduced before the acronyms already given “A, B, C and D” in page 6, line 229.
- Page 7, line 242: “…ratio).The …”, leave a space after the point “…ratio). The …”.
- Page 8, Figure 2: The writing-texts of the images are not clear and make some difficulty to the readers for understanding it. To enlarge further.
In Figure 3, Legend part:
Page 9, Figure 3, legend part: For convenience with curves given, please invert the following acronyms Vc with TFL. Same remark for Page 10, Figure 4 (legend part).
- What do you mean by Vc (Page 9, line 298, Figure 3 and Table 3; page 10, line 311 and Figure 4)? Is it ascorbic acid i.e. vitamin C? . In Materials and Methods part (page 3, line 128) you have mentioned “ascorbic acid” but not “vitamin C”. This makes it very difficult to understand your text. Please, give an appropriate name to Vc in order to avoid the difficulties encountered by the readers.
* Grammatical errors:
- Page 1, line 19: Please revise the following denomination “sulfonica cid” with “sulfonic acid”
- Page 1, line 19: “in the food pharmaceutical…”. Please correct with “in the food and pharmaceutical…”
- Page 1, line 18: Please, change the alphabets (a, a- and b) by the following numbers (2,2- and 1-) in the nomenclature of the DPPH i.e. ‘‘2,2-diphenyl 1-picrylhydrazyl’’ instead of ‘‘a,a-diphenyl b-picrylhydrazyl’’.
Author Response
Dear Professor:
Thank you very much for your responsibility and kindness! I have already carefully read your opinions and made revisions. The details are as follows:
Open Review 1
In Abstract part:
- Page 1, line 14: to develop the acronym UAE i.e. ultrasound-assisted extraction.
A: Accepted. Please see Page 1, line 19.
- Page 1, line 18: Please, change the alphabets (a, a- and b) by the following numbers (2,2- and 1-) in the nomenclature of the DPPH i.e. ‘‘2,2-diphenyl 1-picrylhydrazyl’’ instead of ‘‘a,a-diphenyl b-picrylhydrazyl’’.
A: Accepted. Please see Page 1, line 23.
- Page 1, line 17: What do you means with “TFL”? Is it “TFL” or “TFC : Total Flavonoids Content”. The same questions related to page 2, lines 79 and 87; page 3, lines 96, 104, 109, 119, 122, 124, 134; page 4, lines 144, 157, 179, 182, 184, 186; and in page 5 to page 11.
A: Accepted. Have replaced the word “TFL” with “TFs”. I guess the word “TFs” is much better.
- Page 1, line 24 (keywords): Please add “ultrasound-assisted extraction”.
A: Accepted.
In manuscript text:
- Page 2, line 46: What do you means with “…rom…” ? In order to keep your text homogeneous, please discard this word.
A: Sorry about that. It should be the word “from”.
- Page 2, line 76: What do you means with “TFY”? To develop the acronym given.
A: Have replaced the word “TFY” with “TFs”
- Page 3, line 123: “…by Farias with…” change it with “…by Farias et al. 2020 with…”
A: Accepted.
- Page 3, lines 131, 132 and 133: Respect the writing of the following indices related to As, Ar and A0 i.e. “…As, …Ar and A0…”, they must be conform to the equation (2) given in page 3, line 130.
A: Accepted.
- Page 3, line 137: “…from Lou …[32]”. The author's name “Lou” does not comply with the reference [32] given in page 13. To adjust
A: Have revised.
- Page 4, lines 150 and 151: Respect the writing of the following indices relating to As, Ar and A0 i.e. “…As, …Ar and A0…”, they must be conform to the equation (3) given in page 4, line 149.
A: Have revised.
- Page 6, line 230: For more clarity to readers, the following acronym “Y …” must be introduced before the acronyms already given “A, B, C and D” in page 6, line 229.
A: Have revised.
- Page 7, line 242: “…ratio).The …”, leave a space after the point “…ratio). The …”.
A: Have revised.
- Page 8, Figure 2: The writing-texts of the images are not clear and make some difficulty to the readers for understanding it. To enlarge further.
A: Have revised.
In Figure 3, Legend part:
Page 9, Figure 3, legend part: For convenience with curves given, please invert the following acronyms Vc with TFL. Same remark for Page 10, Figure 4 (legend part).
A: Have revised.
- What do you mean by Vc (Page 9, line 298, Figure 3 and Table 3; page 10, line 311 and Figure 4)? Is it ascorbic acid i.e. vitamin C? . In Materials and Methods part (page 3, line 128) you have mentioned “ascorbic acid” but not “vitamin C”. This makes it very difficult to understand your text. Please, give an appropriate name to Vc in order to avoid the difficulties encountered by the readers.
A: Accepted. The word has been unified as “vitamin C ”
* Grammatical errors:
- Page 1, line 19: Please revise the following denomination “sulfonica cid” with “sulfonic acid”
A: Sorry about that mistake and have revised.
- Page 1, line 19: “in the food pharmaceutical…”. Please correct with “in the food and pharmaceutical…”
A: Sorry about that mistake and have revised.
- Page 1, line 18: Please, change the alphabets (a, a- and b) by the following numbers (2,2- and 1-) in the nomenclature of the DPPH i.e. ‘‘2,2-diphenyl 1-picrylhydrazyl’’ instead of ‘‘a,a-diphenyl b-picrylhydrazyl’’.
A: Have revised.

Reviewer 2 Report
- The title of the manuscript is: "Optimization of Extraction Process, Antioxidant and Antibacterial Capacity of Total Flavonoids from the Flowers of Xanthoceras Sorbifolia Bunge". This title is not clear and should be revised. I suggest the following revised title: "Optimization of flavonoid extraction from Xanthoceras Sorbifolia Bunge Flowers, and the Antioxidant and Antibacterial Capacity of the extract".
- In the background section, the information given for scientific studies of X. Sorbifolia is not sufficient. The discussion should be expended to give more information of results obtained in these studies.
- Line 38-40: this sentence should be revised for the definition of flavonoids: The original sentence is "Flavonoid is a kind of natural compounds with a 2-phenylchromone structure and it is an important secondary class metabolite with low molecular weight that is found in plants [9, 10]. ". This sentence is confusing. Flavonoids are a group of secondary metabolites founds in many plants. IT is unclear here if you are referring to a specific flavonoid or to the flavonoid chemical group. The problem here might be scientific or linguistic. This needs to be clarified.
- The sentence in line 46 is not clear do you mean from? rather then rom?
- Line 32: the term "there are a lot of.." is to casual and should be replaced. A possible re-writing of this sentence is: "there is a range of…".
- Section 2.2: Information is needed as to how long were the ground flowers stored for prior to analysis. Also, what were the storage conditions? Temp? dark vs. light?
- The number of abbreviation s should be reduced when possible. It makes the text hard to read. For example BBD is only sued 3 times throughout the manuscript I see no reason to use an abbreviation for such a small number of repetitions. The authors check the manuscript throughout for the ability to reduce the number of abbreviations.
- Figure 1: the legend should be corrected to include also information about what are the bars. And statistical analyses should be added.
- Table 1: what is RUN in column one. This is not clear.
- Section 3.2.1. It is not clear why this analysis was needed. Why not to conduct a one-way ANOVA for the data of each parameter, followed by a post-hoc test. This would have made the interpretation of the results much easier by allowing the significant letters to be presented in the sub-figures on Fig. 1. If there is a statistical justification of the presented analysis, it should be justified in the text.
- Table 2: it is difficult to follow the results because in the first column to the left the authors use a lettering system without the full name of the parameter (which appears in the figure legend). These should be spelled in full in the Colum, instead or in addition the model letters.
- Figure 3: the legend should be revised to include the number of replications, the statistical analysis conducted etc.
- Figure 4: the legend should be revised to include the number of replications, the statistical analysis conducted etc.
- 3 and figure 4 should be combined to one figure with 2 sub-figures.
Author Response
Dear Professor:
Thank you very much for your responsibility and kindness! I have already carefully read your opinions and made revisions. The details are as follows:
Open Review 2
Comments and Suggestions for Authors
- The title of the manuscript is: "Optimization of Extraction Process, Antioxidant and Antibacterial Capacity of Total Flavonoids from the Flowers of Xanthoceras Sorbifolia Bunge". This title is not clear and should be revised. I suggest the following revised title: "Optimization of flavonoid extraction from Xanthoceras Sorbifolia Bunge Flowers, and the Antioxidant and Antibacterial Capacity of the extract".
A:Thanks for the excellent advice and have accepted it.
- In the background section, the information given for scientific studies of X. Sorbifolia is not sufficient. The discussion should be expended to give more information of results obtained in these studies.
A: Have revised.
- Line 38-40: this sentence should be revised for the definition of flavonoids: The original sentence is "Flavonoid is a kind of natural compounds with a 2-phenylchromone structure and it is an important secondary class metabolite with low molecular weight that is found in plants [9, 10]. ". This sentence is confusing. Flavonoids are a group of secondary metabolites founds in many plants. IT is unclear here if you are referring to a specific flavonoid or to the flavonoid chemical group. The problem here might be scientific or linguistic. This needs to be clarified.
A: Have revised.
- The sentence in line 46 is not clear do you mean from? rather then rom?
A: Have revised.
- Line 32: the term "there are a lot of.." is to casual and should be replaced. A possible re-writing of this sentence is: "there is a range of…".
A: Have revised.
- Section 2.2: Information is needed as to how long were the ground flowers stored for prior to analysis. Also, what were the storage conditions? Temp? dark vs. light?
A: Have revised.
- The number of abbreviation s should be reduced when possible. It makes the text hard to read. For example BBD is only sued 3 times throughout the manuscript I see no reason to use an abbreviation for such a small number of repetitions. The authors check the manuscript throughout for the ability to reduce the number of abbreviations.
A: Have revised.
- Figure 1: the legend should be corrected to include also information about what are the bars. And statistical analyses should be added.
A: Have revised.
- Table 1: what is RUN in column one. This is not clear.
A: Have replaced the word “RUN” with “ NO.”.
- Section 3.2.1. It is not clear why this analysis was needed. Why not to conduct a one-way ANOVA for the data of each parameter, followed by a post-hoc test. This would have made the interpretation of the results much easier by allowing the significant letters to be presented in the sub-figures on Fig. 1. If there is a statistical justification of the presented analysis, it should be justified in the text.
A:This method of ANOVA is commonly used in BBD design, which is conducted according to the equation of second order polynomial fitting.
- Table 2: it is difficult to follow the results because in the first column to the left the authors use a lettering system without the full name of the parameter (which appears in the figure legend). These should be spelled in full in the Colum, instead or in addition the model letters.
A: Because of second order polynomial fitting, the fitting equation also includes quadratic terms ,such as A2, B2, AB and so on. Thus, for the sake of fluency, the full name of the parameter was not used.
- Figure 3: the legend should be revised to include the number of replications, the statistical analysis conducted etc.
A: Have revised.
- Figure 4: the legend should be revised to include the number of replications, the statistical analysis conducted etc.
A: Have revised.
- 3 and figure 4 should be combined to one figure with 2 sub-figures.
A:Figure 3 is in the section 3.3.1 and figure 4 is in the section 3.3.2, so for ease of reading, the two figures are presented separately.
